# Structural evolution of a DNA repair self-resistance mechanism targeting genotoxic secondary metabolites

Elwood A. Mullins [1,5], Jonathan Dorival [1,5], Gong-Li Tang [2], Dale L. Boger [3] & Brandt F. Eichman [1,4✉]

Microbes produce a broad spectrum of antibiotic natural products, including many DNA-damaging genotoxins. Among the most potent of these are DNA alkylating agents in the spirocyclopropylcyclohexadienone (SCPCHD) family, which includes the duocarmycins, CC-1065, gilvusmycin, and yatakemycin. The yatakemycin biosynthesis cluster in *Streptomyces* sp. TP-A0356 contains an AlkD-related DNA glycosylase, YtkR2, that serves as a self-resistance mechanism against yatakemycin toxicity. We previously reported that AlkD, which is not present in an SCPCHD producer, provides only limited resistance against yatakemycin. We now show that YtkR2 and C10R5, a previously uncharacterized homolog found in the CC-1065 biosynthetic gene cluster of *Streptomyces zelensis*, confer far greater resistance against their respective SCPCHD natural products. We identify a structural basis for substrate specificity across gene clusters and show a correlation between in vivo resistance and in vitro enzymatic activity indicating that reduced product affinity—not enhanced substrate recognition—is the evolutionary outcome of selective pressure to provide self-resistance against yatakemycin and CC-1065.

[1] Department of Biological Sciences, Vanderbilt University, Nashville, Tennessee, USA. [2] State Key Laboratory of Bio-organic and Natural Products Chemistry, Shanghai Institute of Organic Chemistry, Chinese Academy of Sciences, Shanghai, China. [3] Department of Chemistry, The Scripps Research Institute, La Jolla, California, USA. [4] Department of Biochemistry, Vanderbilt University School of Medicine, Nashville, Tennessee, USA. [5]These authors contributed equally: Elwood A. Mullins, Jonathan Dorival. ✉email: brandt.eichman@vanderbilt.edu

**B**acteria, fungi, and plants often produce toxic secondary metabolites as defense mechanisms against invading organisms. Streptomycetes produce a staggering number and variety of natural products, including genotoxic DNA-damaging antibiotics[1]. These agents may covalently alkylate or non-covalently intercalate the DNA nucleobases, forming mutagenic and/or toxic adducts that inhibit replication, transcription, or chromosome maturation, leading to chromosome rearrangement and instability, cell death, aging, and disease[2–7]. Among the most potent genotoxic secondary metabolites are those of the spirocyclopropylcyclohexadienone (SCPCHD) family, which includes yatakemycin (YTM), gilvusmycin, CC-1065, and duocarmycin A and SA (DSA) (Fig. 1a)—all of which exhibit high cytotoxicity against tumor cells and pathogenic fungi[8–16]. These molecules bind in the minor groove of AT-rich regions of DNA and undergo binding-induced activation to modify N3 of adenine[17] (Fig. 1b). Despite forming a covalent link with only one strand of DNA, SCPCHD adducts create an extended network of non-covalent CH-π interactions with both strands that greatly stabilizes the DNA duplex and inhibits nucleic acid metabolism[18,19].

Antibiotic-producing microbes have resistance mechanisms against their own toxins that are often genetically clustered with the antibiotic synthesis operon[20–22]. In addition to toxin inactivation and efflux, DNA repair of the toxin-DNA adduct has been identified as a self-resistance mechanism against genotoxic natural products[14,23–32]. Nucleotide excision repair (NER) and base excision repair (BER) pathways serve to eliminate the majority of alkyl-DNA lesions. The NER pathway removes bulky, DNA distorting/destabilizing lesions, and is initiated by an excinuclease complex that isolates the damaged segment of DNA via dual incisions flanking the lesion[33,34]. By contrast, the BER pathway removes small nucleobase adducts, and is initiated by lesion-specific DNA glycosylases that hydrolyze the *N*-glycosidic bond to liberate the aberrant nucleobase from the DNA backbone[35,36]. The resulting apurinic/apyrimidinic (AP) site is incised by an AP endonuclease, generating a terminal 3'-hydroxyl group needed for polymerase-dependent synthesis of new DNA.

Owing to their exquisite helix-stabilizing properties, SCPCHD adducts are poorly repaired by the NER pathway, making these compounds highly toxic with strong antibiotic and antitumor potential[37–40]. To counter this toxicity, at least two of the Streptomycetes that produce SCPCHD natural products encode DNA glycosylases within the corresponding biosynthetic gene clusters[26,29] (Fig. 1c). One of these homologs, YtkR2, has been shown to excise YTM-adenine (YTM-Ade) adducts in vitro and to confer self-resistance against YTM in vivo[26] (Fig. 1b). The other homolog, C10R5, has not been characterized, but seems likely to provide similar self-resistance against CC-1065[29]. YtkR2 and C10R5 are homologs of *Bacillus cereus* AlkD, a unique DNA glycosylase able to excise bulky lesions by virtue of not using the base-flipping mechanism employed by other glycosylases[41–45]. Unlike YtkR2 and CC-1065, AlkD is present in non-antibiotic-producing species of *Bacillus* and provides only limited cellular resistance against YTM[18].

Although YtkR2 has been shown to provide resistance against YTM, the corresponding resistance provided by C10R5 against CC-1065 is unknown. Moreover, the utility of the non-base-flipping mechanism employed by these enzymes to recognize SCPCHD adducts has not been explored outside YTM and AlkD. To elucidate how YtkR2 and C10R5 evolved to efficiently eliminate lesions generated by diverse SCPCHD natural products, we performed a comparative biochemical, biophysical, and structural analysis to identify differences between AlkD and these specialized homologs that are responsible for conferring enhanced cellular resistance against YTM, CC-1065, and DSA. Our data indicate that reduced product affinity,

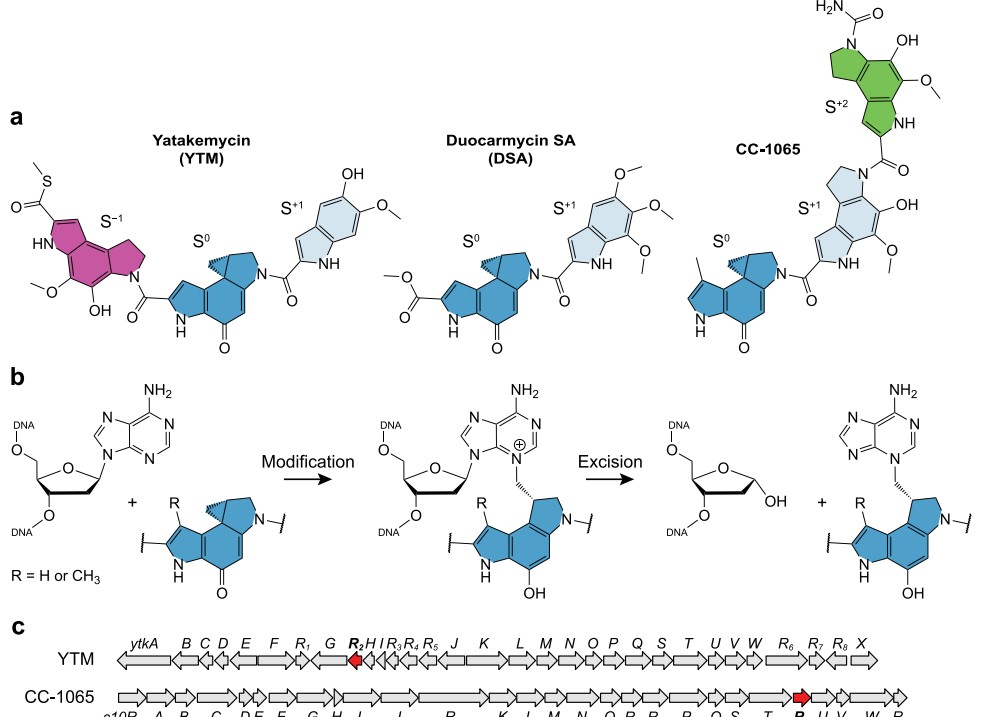

**Fig. 1 DNA modification by the spirocyclopropylcyclohexadienone (SCPCHD) family of natural products. a** Chemical structures of yatakemycin (YTM), duocarmycin SA (DSA), and CC-1065. All three compounds share the $S^0$ subunit (dark cyan), which includes the reactive SCPCHD moiety, and the $S^{+1}$ subunit (pale cyan). YTM and CC-1065 also possess the additional subunits $S^{-1}$ (magenta) and $S^{+2}$ (green), respectively, on either side of the $S^0$ and $S^{+1}$ subunits. **b** Generalized mechanisms of DNA modification by SCPCHD compounds and nucleobase excision by associated DNA glycosylases. **c** *ytk* and *c10* gene clusters responsible for biosynthesis of YTM and CC-1065, respectively. Genes encoding the DNA glycosylases YtkR2 and C10R5 are colored red.

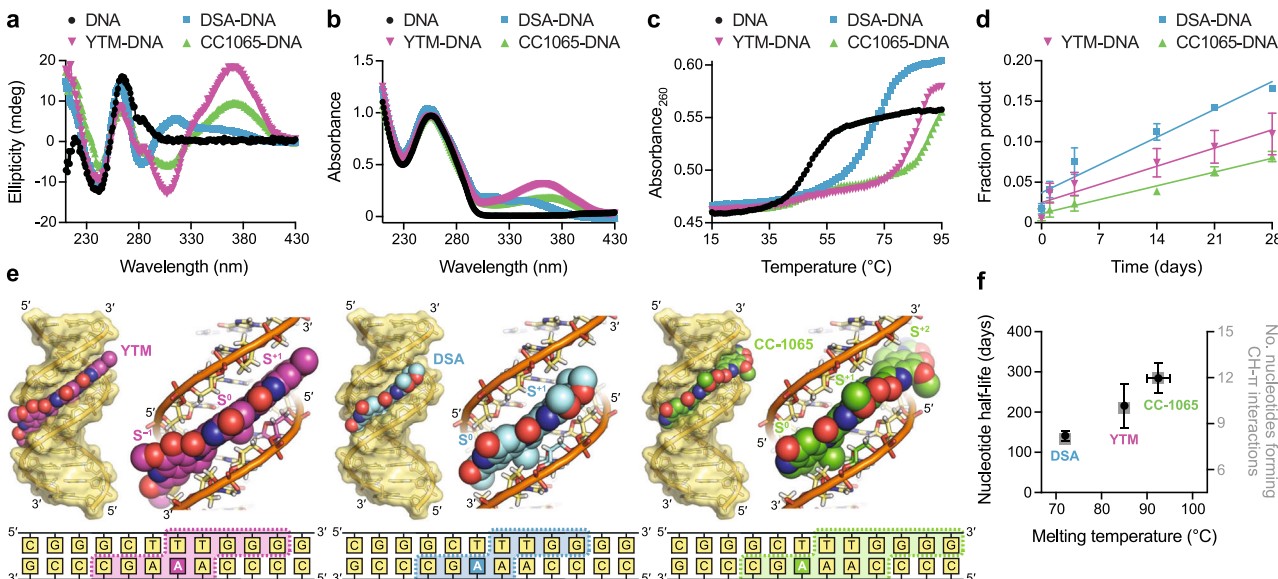

**Fig. 2 Characterization of DNA adducts produced by YTM, DSA, and CC-1065. a, b** Circular dichroism (**a**) and absorbance (**b**) spectra of unmodified and alkylated DNA collected at 15 °C. **c** Thermal melting profiles of unmodified and alkylated DNA. Melting temperatures ($T_{m,DNA}$ = 49 °C, $T_{m,DSA-DNA}$ = 72 °C, $T_{m,YTM-DNA}$ = 85 °C, and $T_{m,CC1065-DNA}$ ≥ 90 °C) were determined by fitting the data to a polynomial function and calculating the inflexion point of the curve. Spectra and melting profiles for unmodified DNA and YTM-DNA were previously described by Mullins et al.[18]. **d** Spontaneous depurination of alkylated DNA at 30 °C. The data were fit to a linear model and half-lives ($t_{1/2,DSA-DNA}$ = 142 ± 11 days, $t_{1/2,YTM-DNA}$ = 224 ± 56 days, and $t_{1/2,CC1065-DNA}$ = 288 ± 36 days) were calculated assuming first-order kinetics. Data are presented as the mean ± SD from three replicate experiments. **e** Hypothetical models of DNA modified by YTM, DSA, and CC-1065. The dashed boxes indicate nucleotides that are predicted to form CH-π interactions with the SCPCHD adducts. **f** Correlation between melting temperature, rate of spontaneous depurination (plotted as nucleotide half-life), and number of nucleotides forming CH-π interactions, derived from **c**, **d**, and **e**, respectively. Horizontal error bars represent uncertainty in the melting temperature of CC1065-DNA caused by the lack of an endpoint in the melting profile. As such, the estimated melting temperature of CC1065-DNA was plotted as the average value ± the range of values calculated assuming endpoints between those of unmodified DNA (minimum) and DSA-DNA (maximum). Nucleotide half-lives are presented as the mean ± SD from three replicate experiments. Source data for **a**–**d** and **f** are provided as a Source Data file.

not enhanced substrate recognition, is the key factor associated with improved cellular resistance.

## Results

**SCPCHD adducts stabilize the DNA duplex through a network of CH-π interactions**. The five known SCPCHD natural products can be divided into three sub-types based on the number and arrangement of the indole and pyrroloindole subunits (Fig. 1a). To better understand how these differences affect toxicity and repair, we selected one compound from each group for biochemical and biophysical characterization (Fig. 2). We previously demonstrated that a single YTM adduct in a short oligonucleotide retains a B-DNA conformation and dramatically increases the stability of the duplex, providing an explanation for why these lesions are poorly repaired by the NER pathway[18,46]. Similarly, earlier work showed that multiple CC-1065 adducts greatly increase the melting temperature of genomic DNA[19]. To directly compare the effects of the different SCPCHD sub-types on DNA stability and structure, we introduced a single YTM, DSA, or CC-1065 adduct onto a central adenine within the same GC-rich DNA dodecamer. The circular dichroism spectra of each construct showed negative and positive bands at ~240 and 270 nm, respectively, indicative of B-form DNA[47] (Fig. 2a). Additional features in the spectra likely result from either minor structural perturbation of the DNA or dichroism of the SCPCHD adducts themselves, which produce broad peaks above 300 nm in the absorbance spectra (Fig. 2b). To assess the effect of each compound on duplex stability, thermal denaturation was monitored by an increase in absorbance at 260 nm (Fig. 2c). Relative to unmodified DNA, the presence of a single DSA, YTM, or CC-1065 adduct increased the melting temperature by 23, 36, or ≥41 °C, respectively.

The increase in duplex stability correlated with the stability of the adduct. After 28 days at 30 °C, only 16%, 11%, and 8%, respectively, of DSA, YTM, and CC-1065 adducts had spontaneously depurinated (Fig. 2d and Supplementary Fig. 1). For comparison, the half-life of 3-methyl-2′-deoxyadenosine in genomic DNA is <4 days under similar conditions (interpolated from previously published rates at 22 °C and 39 °C)[48].

To elucidate the structural and chemical origins of these trends, we modeled each of the constructs by manually docking ideal B-form DNA and SCPCHD adducts taken from crystallographic AlkD product complexes (described below) (Fig. 2e). Despite being maintained as rigid bodies, all adducts fit in the minor groove without steric clashes. In these hypothetical models, the hydrophilic edges of the subunits are exposed to solvent, whereas the hydrophobic faces are sandwiched between the backbones of the two DNA strands. As we previously described with YTM adducts[18], this arrangement creates an extended network of CH-π interactions. Although these interactions are individually often weak (~1 kcal/mol)[49], the combined strength of many such interactions generates a strong energy barrier to duplex denaturation and lesion depurination. Indeed, DNA stability and lesion half-life correlate with the number of nucleotides that form CH-π interactions (Fig. 2f). Thus, this network of weak interactions explains the trends in duplex stability and nucleotide half-life that we observe with each adduct. Importantly, our models and conclusions are consistent with a previously published NMR structure of DSA-DNA (PDB accession 1DSA)[50]. (Supplementary Fig. 2), which further highlights the importance of CH-π interactions on the stability of SCPCHD-modified DNA duplexes. Relative to ideal B-form DNA, the DSA adduct in

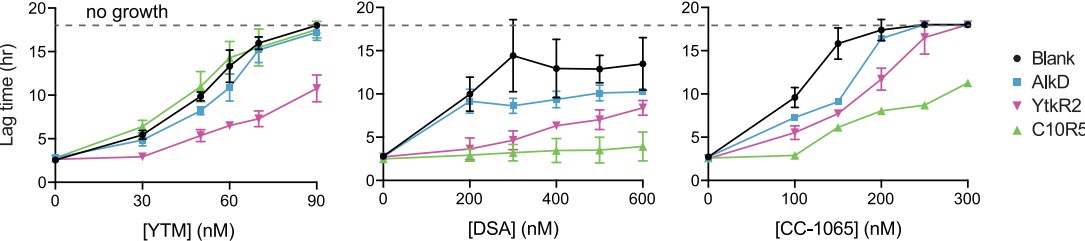

**Fig. 3 Determination of drug resistance.** *E. coli* BL21(DE3) cells transformed with either an empty plasmid (blank) or a plasmid encoding AlkD, YtkR2, or C10R5 were grown in the presence of various concentrations of YTM, DSA, or CC-1065. A shorter lag phase (lag time) before the onset of exponential growth indicates increased drug resistance. A complete lack of cell growth was plotted as a lag time of 18 h, the total incubation period for the cultures, as indicated by the dashed line. Data are presented as the mean ± SD from four replicate experiments. Source data are provided as a Source Data file.

the experimental structure induces a slight narrowing of the minor groove and twisting of the DNA backbone, which maximizes the number and strength of the CH-π interactions formed between the adduct and the DNA. Thus, through a network of CH-π interactions, SCPCHD adducts create an induced fit in the minor groove that tethers the opposing DNA strands and prevents gross deformation to the B-DNA conformation, explaining how these adducts pose a significant challenge to repair by the UvrABC NER excinuclease[18].

**YtkR2 and C10R5 evolved to provide self-resistance against YTM and CC-1065.** Historically, bulky DNA lesions, such as SCPCHD adducts, have been regarded as substrates for the NER pathway[51–54]. However, we previously showed that NER provides *Bacillus anthracis* with only limited resistance against YTM, while AlkD-mediated BER provides a modest degree of additional resistance[18]. At the time, we speculated the AlkD homologs that evolved in antibiotic-producing bacteria would provide greater resistance against their SCPCHD natural products. To test this theory, we transformed *Escherichia coli* with either an empty plasmid or a plasmid encoding AlkD, YtkR2, or C10R5, and incubated the cultures with various concentrations of YTM, DSA, or CC-1065 (Supplementary Fig. 3). Sensitivity to the compounds was scored as the lag time, or delay prior to exponential cell growth[55,56] (Fig. 3). In the absence of any drug, cell growth was unchanged regardless of the protein being expressed. Conversely, cells transformed with only an empty vector showed different degrees of resistance against the three SCPCHD compounds. As previously observed with eukaryotic cells[13], YTM was the most toxic, inhibiting growth at the lowest drug concentration, followed by CC-1065 and then DSA. In cultures containing YTM, only YtkR2 provided clear and significant resistance, shortening the lag phase before exponential growth. Similarly, in cultures containing CC-1065, C10R5 provided the greatest resistance, although YtkR2, and to a much lesser extent AlkD, also provided additional resistance. The same trend was observed in cultures containing DSA; C10R5 provided the greatest resistance, followed by YtkR2 and then AlkD. These data are consistent with the notion that YtkR2 and C10R5 evolved to provide self-resistance against YTM and CC-1065, respectively. However, both proteins also confer substantial resistance against DSA, and YtkR2 provides moderate resistance against CC-1065. This lack of specificity suggests that the features of YtkR2 and C10R5 responsible for drug resistance are largely independent of the differences between the SCPCHD compounds. The only exception to this generalization appears to be the lack of resistance against YTM provided by C10R5.

**Product affinity inversely correlates with cellular resistance**. To understand how YtkR2 and C10R5 confer greater resistance than AlkD to SCPCHD natural products, we measured in vitro exci-

sion activity using purified proteins and defined oligonucleotide substrates (Fig. 4a,b and Supplementary Fig. 4). Under single-turnover conditions, in which the enzymes are in molar excess of the DNA substrates, both AlkD and YtkR2 rapidly excised all three adducts, approaching the endpoints of the reactions within 15 s (Fig. 4c, Supplementary Fig. 5, and Supplementary Table 1). C10R5 rapidly excised DSA and CC-1065 lesions, albeit more slowly than AlkD and YtkR2, but removed YTM adducts at a substantially reduced rate. Under multiple-turnover conditions, in which the DNA substrates are in molar excess of the enzymes, the results were strikingly different (Fig. 4d, Supplementary Fig. 6, and Supplementary Table 1). Although AlkD rapidly excised a molar equivalent of each adduct (i.e., the first turnover in which ~10% product is formed), subsequent turnovers occurred several orders of magnitude more slowly. Excision of adducts by YtkR2 was similarly slowed, although to a lesser extent. Surprisingly, the multiple-turnover rates of removal of DSA and CC-1065 lesions by C10R5 were effectively unchanged; only the rate of excision of YTM adducts was substantially reduced. Thus, under single-turnover conditions, all homologs have the capacity for rapid excision of diverse SCPCHD lesions, although this capacity appears to be reduced in C10R5, and inefficient substrate recognition by C10R5 seemingly further slows removal of YTM lesions. For all homologs, there is only weak correlation between rates of single-turnover excision and cellular resistance (Figs. 3 and 4c, and Supplementary Table 1). Conversely, the multiple-turnover rates of the three enzymes are strongly correlated with their cellular resistance against each SCPCHD compound (Figs. 3 and 4d, and Supplementary Table 1).

We previously demonstrated that slow multiple turnover by AlkD was correlated with inhibition of AP endonuclease activity in vitro, which we attributed to tight binding of the AP product generated upon base excision[18,57]. The substantially faster multiple turnover that we observed with YtkR2 and C10R5 therefore suggests that product affinity is weaker for these homologs. To test this possibility, we performed AP-DNA incision assays using the AP endonuclease EndoIV and either AP-DNA generated from each of the SCPCHD adducts or tetrahydrofuran (THF)-DNA containing an AP analog (Fig. 4e and Supplementary Fig. 7). Reaction mixtures with THF-DNA were supplemented with free 3-methyladenine (3mAde) nucleobase to approximate the nucleobase adducts excised from the SCPCHD lesions, while also providing a basis to assess the effects of the excised SCPCHD adducts on product affinity. Relative to AP-DNA produced by AlkD, AP-DNA generated by YtkR2 or C10R5 was incised by EndoIV to a substantially greater extent, regardless of which SCPCHD adduct was excised to produce the abasic site, consistent with relatively weak product binding by YtkR2 and C10R5. For each of the homologs, AP-DNA generated from DSA-DNA was cleaved to the greatest extent, which could be explained by DSA-Ade being bound by each of the homologs

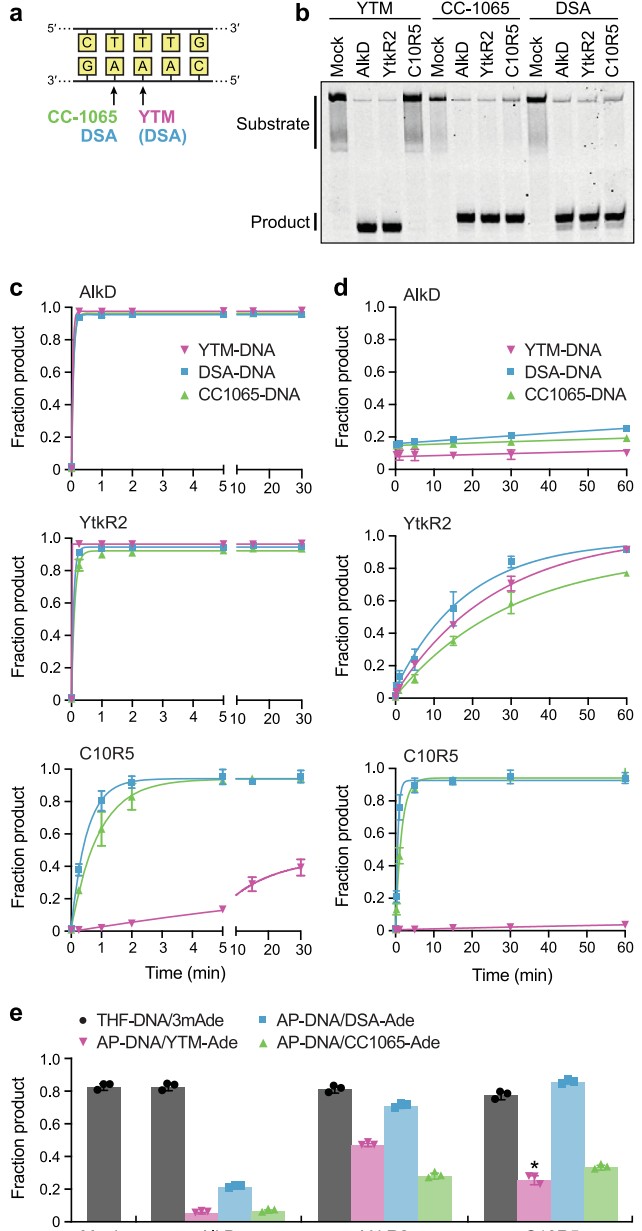

**Fig. 4 Measurement of adduct excision activity. a** Sequence specificity of SCPCHD natural products. Modified nucleotides are indicated with arrows. The central 2'-deoxyadenosine nucleotide is modified by DSA with relatively low frequency. **b** Separation of full-length DNA substrate (25-mer) from alkali-cleaved product (12- or 13-mer) by denaturing gel electrophoresis. Oligonucleotide substrates were incubated with or without (mock) enzyme for 30 s before reactions were quenched with sodium hydroxide and heat to denature both the enzyme and the DNA and to cleave the DNA at the AP site. The high melting temperature of the GC-rich substrate prevented complete denaturation of the DNA duplex, producing a smeared substrate band. The unique sequence specificities of YTM, CC-1065, and DSA for different 2'-deoxyadenosine nucleotides result in generation of alkali-cleaved products of different lengths. The lower specificity of DSA allows for modification of one of two nucleotides, each corresponding to a different product band. Experiments were performed in triplicate. **c** Single-turnover excision of SCPCHD adducts. Reactions contained 1 µM enzyme and 100 nM DNA. With all substrates, both AlkD and YtkR2 approached the endpoints of the reactions by the first time point, potentially reducing or masking differences in excision rates between the adducts and between the enzymes. **d** Multiple-turnover excision of SCPCHD adducts. Reactions contained 10 nM enzyme and 100 nM DNA. **e** Incision of AP-DNA. SCPCHD adducts were pre-incubated with AlkD, YtkR2, or C10R5 to generate AP-DNA for subsequent incision by EndoIV. THF-DNA and 3mAde nucleobase were also pre-incubated with each homolog or without enzyme (mock). Incomplete excision of YTM-Ade by C10R5 during the pre-incubation reaction reduced the amount of AP-DNA (Supplementary Fig. 7), limiting the possible fraction product. The affected quantity is indicated with an asterisk. Of the AP-DNA produced, 98% was subsequently incised by EndoIV. Data in **c–e** are presented as the mean ± SD from three replicate experiments. Source data are provided as a Source Data file.

**Diverse SCPCHD adducts are recognized primarily through nonspecific interactions.** To understand AlkD's lack of specificity for the three SCPCHD sub-types and comparatively high product affinity, we determined crystal structures of AlkD bound to DNA containing each of the three adducts. We previously reported the structure of AlkD bound to AP-DNA and YTM-Ade (PDB accession 5UUG)[18] and now report structures of AlkD bound to AP-DNA and either DSA-Ade or CC1065-Ade (Fig. 5 and Supplementary Table 2). Product complexes were generated by incubating AlkD with the corresponding DNA substrates, allowing for enzymatic hydrolysis of the *N*-glycosidic bond and formation of an AP site and a free nucleobase adduct (Fig. 5a). The conformations of AlkD and the abasic DNA are effectively identical in the three structures (Fig. 5b and Supplementary Fig. 8). Relative to the substrate models (Fig. 2e), binding by AlkD bends the B-DNA helix by ~30° and widens the minor groove by 4–5 Å, disrupting the CH-π interactions between the adducts and the modified strand, and enabling the three catalytic residues—Trp109, Asp113, and Trp187—to contact the modified nucleotide[18,43,58]. Despite this remodeling, the excised nucleobases remain stacked in the DNA duplex, with the bulky SCPCHD adducts bound within an extended cleft between the concave surface of the protein and the minor groove of the DNA.

Within the SCPCHD binding cleft, 14 protein residues contact one or more of the adducts (Fig. 5c). Only three—Gln38, Lys156, and Lys194—form adduct-specific hydrogen-bonding interactions. Gln38 interacts with the hydroxyl and methoxy substituents of the S[−1] subunit unique to YTM, while Lys194 interacts with the methoxy substituent of the S[+2] subunit unique to CC-1065. Otherwise, Gln38 and Lys194 function as DNA-binding residues. Lys156 contacts all three adducts but only forms a hydrogen-bonding interaction with the hydroxyl substituent

with weaker affinity, producing a less stable ternary product complex, and/or by DSA-Ade inhibiting AP-DNA incision by EndoIV to a lesser degree. The former would be consistent with the fewer observed or predicted protein-adduct binding interactions formed with DSA-Ade (described below), whereas the latter would be consistent with the fewer predicted CH-π interactions formed with DSA (Fig. 2e, f). Unlike AP-DNA, the extent to which THF-DNA was incised by EndoIV was independent of which DNA glycosylase homolog was present, and no additional THF-DNA was incised in reactions mixtures in which AlkD, YtkR2, and C10R5 were omitted (Fig. 4e and Supplementary Fig. 7), indicating product affinity is greater for abasic DNA in the presence of the excised SCPCHD adducts. Taken together, the base excision and AP-DNA incision data suggest that YtkR2 and C10R5 evolved to possess lower product affinities than AlkD for abasic DNA derived from SCPCHD lesions. Moreover, this apparently reduced product affinity correlates with the increased cellular resistance provided by these specialized homologs (Figs. 3 and 4d,e).

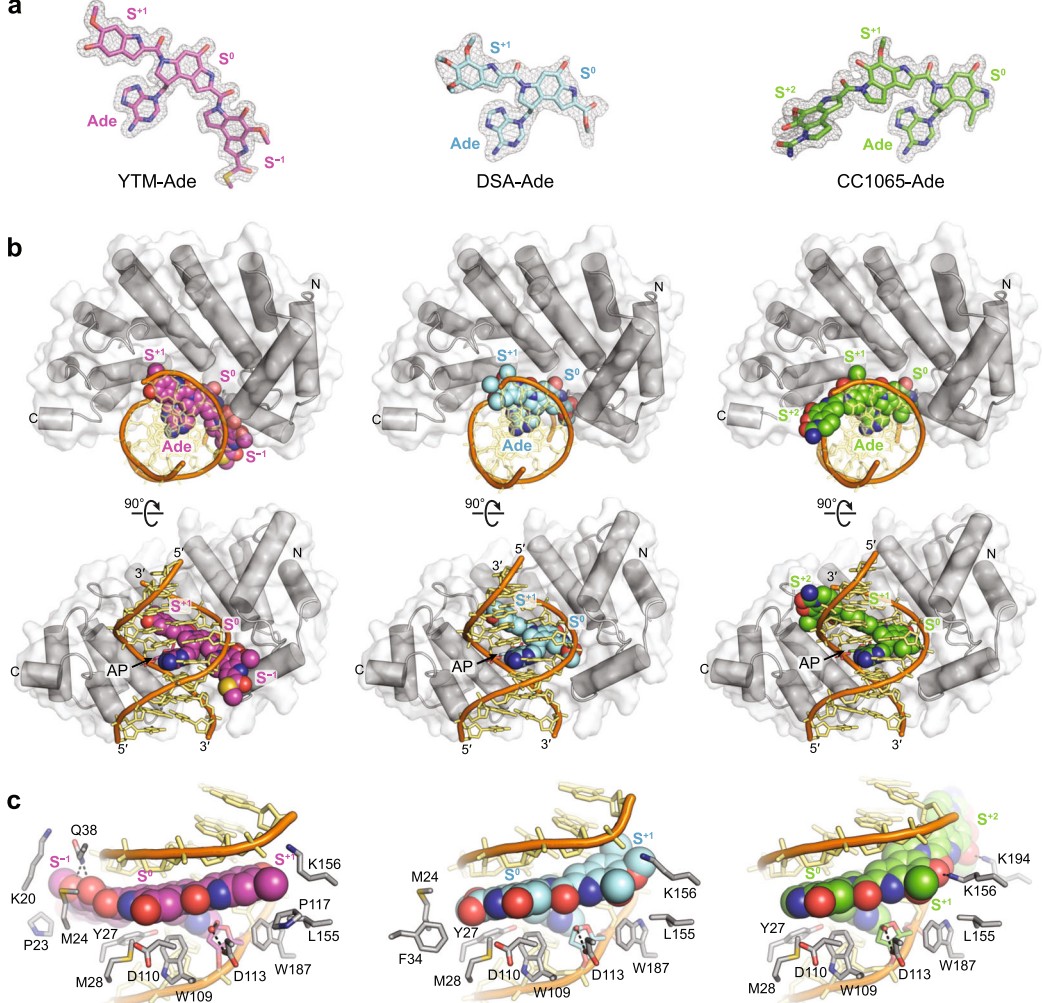

**Fig. 5 Comparison of AlkD product complexes. a** Excised SCPCHD adducts from the crystallographic AlkD product complexes. Annealed omit $mF_o - DF_c$ electron density maps are contoured to $3\sigma$ and carved around the omitted atoms with a 2 Å radius. **b** Orthogonal views of AlkD in complex with AP-DNA and either YTM-Ade (magenta), DSA-Ade (cyan), or CC1065-Ade (green). **c** Close-up views of the AlkD product complexes showing all residues that interact with the SCPCHD adducts. Hydrogen-bonding interactions are indicated with dashed lines. The structure of AlkD bound to AP-DNA and YTM-Ade (PDB accession 5UUG) was previously described by Mullins et al.[18].

specific to the $S^{+1}$ subunit of CC-1065. In YTM and DSA, this position is occupied by a methoxy group, forcing Lys156 to adopt a different rotamer that hydrogen bonds with the DNA backbone. All other hydrogen bonds between the adducts and the protein are mediated by water. Additional recognition of the adducts occurs through hydrophobic contacts and CH-π interactions. The majority of these binding interactions involve the $S^0$ and $S^{+1}$ subunits present in all of the SCPCHD adducts. Of seemingly principal importance are Tyr27, Met28, Trp109, Asp110, Leu155, and Trp187. Together, these residues form a shelf below the adducts and provide an alternative network of CH-π interactions, replacing the network of CH-π interactions that is disrupted upon binding of the DNA by AlkD and widening of the minor groove. Thus, AlkD's substrate-binding cleft can accommodate each SCPCHD compound with strikingly few adduct-specific interactions. This lack of specificity is consistent with AlkD's similar rates of excision for the three substrates.

To explain the structural basis for cellular resistance against SCPCHD compounds provided by YtkR2 and C10R5, the AlkD product complexes were used as templates to create homology models of both homologs bound to abasic DNA and each of the nucleobase adducts (Supplementary Fig. 9). Despite the low

sequence identity (19–25%) among the three proteins, the only putative structural difference of note is an extended loop between helices αI and αJ in both YtkR2 and C10R5. The three homologs appear to share the same three catalytic residues and a high conservation among DNA-binding residues, although YtkR2 and C10R5 possess fewer apparent DNA-binding residues (15 and 14, respectively) than the 18 present in AlkD. Within the substrate-binding cleft, the largest variation predicted by the homology models is in the residues that contact the SCPCHD adducts. However, despite this variation, YtkR2 and C10R5 seemingly form few specific interactions with any of the adducts, including the adduct that each evolved to excise. The only putative hydrogen-bonding interactions between YtkR2 and YTM are formed between Arg42 and the hydroxyl and methoxy substituents of the $S^{-1}$ subunit. Similarly, the only likely hydrogen-bonding interactions between C10R5 and CC-1065 are formed between Lys159, Arg205, and the hydroxyl and methoxy substituents of the $S^{+1}$ and $S^{+2}$ subunits. Surprisingly, all of the adduct-binding residues in YtkR2 and C10R5 that appear to be specific for YTM and CC-1065 are equivalent to the three DNA-binding residues in AlkD that form similar interactions with YTM-Ade and CC1065-Ade (Fig. 5c and Supplementary Fig. 9).

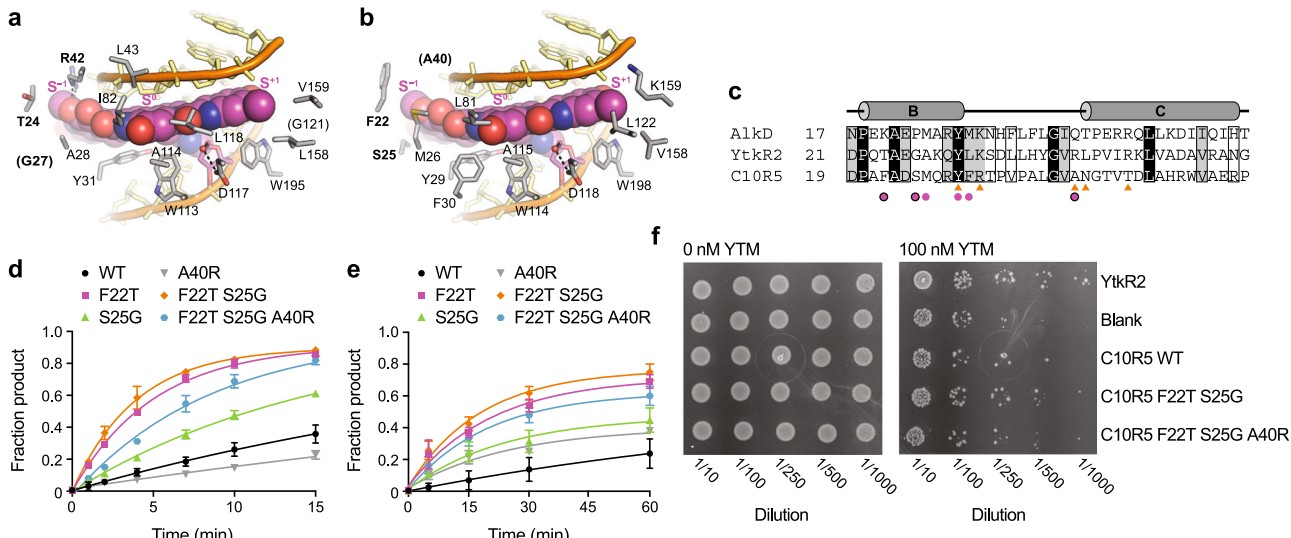

**Fig. 6 Molecular basis of C10R5 substrate specificity. a, b** Close-up views of YtkR2 (**a**) and C10R5 (**b**) homology models bound to AP-DNA and YTM-Ade. Hydrogen-bonding interactions are indicated with dashed lines. **c** Truncated sequence alignment of AlkD, YtkR2, and C10R5. Residues that interact with AP-DNA and YTM-Ade in the AlkD product complex (PDB accession 5UUG) are indicated with orange triangles and magenta circles, respectively. **d** Single-turnover excision of YTM-Ade by wild-type and mutant C10R5. Reactions contained 1 μM enzyme and 100 nM DNA. **e** Multiple-turnover excision of YTM-Ade by wild-type and mutant C10R5. Reactions contained 10 nM enzyme and 100 nM DNA. Mutated residues are indicated with bold font in **a** and **b** and with black circles in **c**. Data in **d** and **e** are presented as the mean ± SD from three replicate experiments. Source data are provided as a Source Data file. **f** YTM resistance conferred by C10R5 mutants. *E. coli* BL21(DE3) cells were transformed with either an empty plasmid (blank) or a plasmid encoding YtkR2, C10R5, or one of two C10R5 mutants, and grown in the absence or presence of YTM. Experiments were performed in duplicate.

As with AlkD, the remainder of the binding interactions are composed of water-mediated hydrogen bonds, hydrophobic contacts, and CH-π interactions. This apparent lack of structural specificity, combined with the nearly complete lack of specificity observed in the excision assays, suggests the enhanced cellular resistance provided by YtkR2 and C10R5 does not result from enhanced recognition of a particular SCPCHD adduct. Instead, the additional cellular resistance provided by YtkR2 and C10R5 seems more likely to result from decreased product affinity, which may be explained, at least in part, by the reduced number of predicted DNA-binding residues in these homologs (Supplementary Fig. 9). It is also possible that other, more subtle, structural factors that are not apparent from the homology models may play a role.

**C10R5 poorly accommodates the unique arrangement of subunits in YTM**. The only significant indication of substrate specificity by any of the three homologs was C10R5's relatively slow excision of YTM-Ade adducts in vitro and corresponding lack of YTM resistance in cells. Given the structural differences between the SCPCHD natural products, we postulated that unfavorable interactions between C10R5 and the $S^{-1}$ subunit unique to YTM were likely responsible for this reduced activity. Comparison of the YtkR2 and C10R5 homology models indicated three residues (Phe22, Ser25, and Ala40) in C10R5 that were likely to contact the $S^{-1}$ subunit and that were not conserved in either YtkR2 or AlkD (Fig. 6a–c and Supplementary Fig. 9). To determine the influence of these residues on substrate specificity, we mutated them to the corresponding residues in YtkR2 and measured YTM-Ade excision under modified single- and multiple-turnover conditions (Fig. 6d,e, Supplementary Figs. 10 and 11, and Supplementary Table 3). Under single-turnover conditions, both the F22T and S25G single mutants exhibited a modest increase in YTM-Ade excision activity relative to wild-type C10R5. This effect was further enhanced in the F22T-S25G double mutant. Moreover, the increased activity of the C10R5 F22T-S25G double mutant toward YTM-Ade was within five-fold of wild-type

C10R5's activity toward CC1065-Ade (Supplementary Tables 1 and 3). In contrast, an A40R single mutant and an F22T-S25G-A40R triple mutant displayed decreased YTM-Ade excision activity relative to the wild-type enzyme and the F22T-S25G double mutant, respectively. Under multiple-turnover conditions, all C10R5 single mutants, including A40R, exhibited increased YTM-Ade excision activity, which was further increased in the F22T-S25G double mutant. As observed under single-turnover conditions, the F22T-S25G-A40R triple mutant displayed reduced YTM-Ade excision activity relative to the double mutant. Despite this reduction, the YTM-Ade excision activity of the triple mutant was comparable to that of YtkR2 under multiple-turnover conditions, whereas the YTM-Ade excision activities of the F22T single mutant and the F22T-S25G double mutant were slightly greater than that of YtkR2 (Supplementary Tables 1 and 3). These comparatively rapid multiple-turnover rates, however, are likely more indicative of C10R5's apparently low product affinity than vastly improved YTM recognition, as the rates at which these mutants removed YTM-Ade is more than 11-fold slower than the rates at which wild-type C10R5 removed CC1065-Ade and DSA-Ade. Moreover, under single-turnover conditions, the YTM-Ade excision activity of YtkR2 vastly exceeded that of all C10R5 mutants. Thus, although the substitutions (F22T and S25G) that increased the volume of the adduct-binding cleft also increased the rate of YTM-Ade excision, suggesting that C10R5's low activity for YTM-Ade is dictated by a sterically constrained binding cleft at the position of the $S^{-1}$ subunit, additional unfavorable interactions may also be hindering recognition of YTM lesions (Supplementary Figs. 8 and 9), preventing more rapid excision of YTM-Ade adducts.

To determine the biological relevance of the gain-of-activity mutations, we transformed *E. coli* with either an empty plasmid or a plasmid encoding YtkR2, C10R5, C10R5 F22T S25G, or C10R5 F22T S25G A40R, and incubated cells on solid medium containing various concentrations of YTM (Fig. 6f and Supplementary Fig. 12). In the absence of YTM, cell growth was unaffected by the protein construct being expressed. In the

presence of YTM, YtkR2 provided the greatest resistance, whereas cells containing an empty plasmid were the most sensitive. As in our growth experiments in liquid medium (Fig. 3 and Supplementary Fig. 3), wild-type C10R5 provided little to no resistance to YTM (Fig. 6f and Supplementary Fig. 12). The C10R5 double mutant, however, provided clear resistance, albeit less than YtkR2. Consistent with our YTM-Ade excision assays (Fig. 6d, e), the C10R5 triple mutant provided less resistance than the double mutant, and only arguably provided more resistance than wild-type C10R5 (Fig. 6f and Supplementary Fig. 12). These relatively modest increases in YTM resistance may be limited by a number of factors, including potentially reduced expression and/ or solubility of the mutant proteins. Regardless, these cellular results validate our structural and biochemical conclusions, showing homology-based mutations that likely only eliminate steric clashes between C10R5 and YTM—without introducing specific adduct-binding interactions—are able to increase the YTM resistance conferred by C10R5.

## Discussion
In our experiments, YtkR2 and C10R5 provided the greatest cellular resistance to YTM and CC-1065 toxicity, respectively, consistent with evolution of these proteins within the context of the biosynthetic gene cluster for each natural product. Both enzymes also conferred substantial resistance against DSA, even though DSA differs from YTM and CC-1065 in the substituents present on the $S^{+1}$ subunit. YtkR2 also provided significant resistance against CC-1065, which was somewhat unexpected because of the presence of the unique $S^{+2}$ subunit, and suggests that interactions with this subunit do not play an important role in determining substrate specificity. Indeed, the outward face of the $S^{+2}$ subunit is largely solvent exposed and has the fewest protein contacts of any subunit. In contrast, the $S^{-1}$ subunit unique to YTM makes several direct protein contacts, which we show largely account for the weak YTM-Ade excision activity of C10R5. Interestingly, the $S^{-1}$ binding surface is formed predominantly from helix αB, which is a critical DNA-binding element within the AlkD family of enzymes[57] and whose structural variability among HLR glycosylases enables binding to a diverse array of ligands[44,59]. Steric accommodation of the $S^{-1}$ subunit is the only determinant of substrate specificity that we were able to identify through our comparative analysis. Within the largely water-filled adduct-binding cleft, most hydrogen bonds are mediated by a network of water molecules, which creates a seemingly malleable hydrogen-bonding interface able to accommodate different substituents on the $S^0$ and $S^{+1}$ subunits. Similarly, most of the protein residues that directly contact the adducts form a network of CH-π interactions that is only selective for aromatic groups, which are present in all of the indole and pyrroloindole subunits in all of the SCPCHD adducts. This mode of substrate recognition seems largely incompatible with a high degree of substrate specificity, which is consistent with our experimental observations.

Despite selective pressure for YtkR2 and C10R5 to provide self-resistance against YTM and CC-1065, neither homolog evolved to remove YTM-Ade or CC1065-Ade adducts at a substantially increased rate, as observed by the strong correlation between multiple-turnover excision and cellular resistance. Instead, selective pressure seemingly induced both enzymes to develop reduced product affinity. We previously showed that AlkD's low level of protection against YTM results from product inhibition in which the enzyme remains bound to the newly formed abasic site in the DNA, slowing multiple-turnover excision, as well as impeding subsequent repair steps in the BER pathway[18]. Both YtkR2 and C10R5 have evolved to circumvent this inhibition, while also minimizing their impediment to later repair steps. Reduced product inhibition may be explained by the fewer DNA-

binding residues predicted in these proteins, which are also the only contacts predicted to form "specific" hydrogen bonds with the adducts. It is interesting to speculate that the evolution of YtkR2 and C10R5 to improve multiple turnover occurred as a means to handle the high concentration of YTM or CC-1065 in the producing strains, whereas the rapid single-turnover activity of AlkD is sufficient to provide resistance to the presumably much lower intracellular quantities of these compounds in *Bacillus* and other non-producing bacteria.

Until recently, all DNA glycosylases were believed to recognize and excise substrates using a base-flipping mechanism in which the target adduct is sequestered from the DNA helix and into a nucleobase-binding pocket on the surface of the enzyme. The AlkD/YtkR2/ C10R5 enzymes belong to one of two known bacterial DNA glycosylase superfamilies that have evolved a non-base-flipping mechanism as a means to remove structurally diverse duplex-stabilizing adducts[18,43,45]. Although their SCPCHD substrates form a covalent bond with only one DNA strand, a network of CH-π interactions non-covalently tethers the two DNA strands together. This greatly stabilizes the duplex, while also hindering the base-flipping activity necessary for recognition and excision by other repair proteins. An unrelated DNA glycosylase, AlkZ, from *Streptomyces sahachiroi* was recently found from its association with the biosynthetic gene cluster for azinomycin B, a highly toxic interstrand crosslinking agent, and shown to provide self-resistance through unhooking of the corresponding DNA crosslinks[27,60]. Enzymes in the AlkZ superfamily have since been shown to remove a variety of alkylated DNA adducts and interstrand crosslinks[27,61,62]. Despite their different polypeptide folds, AlkZ and AlkD family enzymes adopt a similar C-shaped DNA-binding architecture that enables access to the target *N*-glycosidic bond while the aberrant nucleotide remains stacked in the DNA. This mode of recognition and excision not only eliminates the problem of flipping a duplex-stabilizing lesion out of the DNA, but also avoids the steric limitations imposed by a nucleobase-binding pocket. Thus, the expanded substrate spectrum of non-base-flipping DNA glycosylases broadens the biological utility of the BER pathway, providing a mechanism for resistance against the genotoxic natural products generated by many bacteria.

## Methods
**Protein purification**. *B. cereus* AlkD and *Streptomyces* sp. TP-A0356 YtkR2 were purified as previously described[43,63]. The gene encoding *Streptomyces zelensis* C10R5 (GenBank accession KY379149) was synthesized by Genscript and ligated into a modified pET27 expression vector encoding a Rhinovirus 3C-cleavable hexahistidine-SUMO fusion tag. Overexpression of C10R5 was performed in *E. coli* BL21(DE3) cells grown in Luria Bertani (LB) medium supplemented with 30 mg/L kanamycin. Upon reaching mid-log phase, cultures were cooled from 37 °C to 18 °C, and isopropyl β-D-1-thiogalactopyranoside (IPTG) and glucose were added to final concentrations of 0.5 mM and 0.4 g/L, respectively, to initiate slow protein production. After overnight incubation at 18 °C, cells were collected by centrifugation, resuspended in lysis buffer [50 mM sodium phosphate pH 6.8, 250 mM NaCl, 5% (v/v) glycerol, and 5% (w/v) mannitol], and lysed by passage through an Avestin Emulsifier C3 homogenizer operating at ~15,000 psi. Lysate was cleared by centrifugation and then injected onto a Ni-NTA column. The column was washed with lysis buffer supplemented with 40 mM imidazole and the protein was then eluted using the same buffer but containing 300 mM imidazole. Fractions were pooled and the hexahistidine-SUMO tag was removed by overnight cleavage at 4 °C. C10R5 was diluted three-fold in buffer H [25 mM sodium phosphate pH 6.8, 10% (v/v) glycerol, and 1 mM dithiothreitol (DTT)] before being applied to a heparin Sepharose column equilibrated in buffer H supplemented with 100 mM NaCl. The protein was eluted by linearly increasing the concentration of NaCl to 1 M. Pooled fractions were reapplied to the Ni-NTA column and C10R5 was eluted by washing the column with lysis buffer containing 50 mM imidazole. The protein was concentrated by diafiltration, injected onto a Superdex 200 column equilibrated in buffer S [20 mM MOPS pH 6.8, 300 mM NaCl, and 5% (v/v) glycerol], and eluted in the same buffer. Fractions containing C10R5 were pooled and concentrated by diafiltration. Aliquots were frozen in liquid nitrogen and stored at −80 °C. C10R5 mutants were generated using the Q5 site-directed mutagenesis kit (New England Biolabs) and purified in the same manner as the wild-type protein. *E. coli* EndoIV was purchased from New England Biolabs.

**Adduct preparation**. Oligodeoxynucleotides were purchased from Integrated DNA Technologies and used without further purification. YTM and CC-1065 were purified and DSA was synthesized as previously described[14,29,64]. DNA adducts were generated as previously described[18], except reaction mixtures contained 10 mM MES pH 6.5, 40 mM NaCl, 10% (v/v) dimethylsulfoxide, 10 μM DNA, and 150 μM YTM or 100 μM DSA or CC-1065. After 24 h at 22 °C, unreacted DSA, CC-1065, and YTM were removed by passing the reaction mixtures through a Sephadex G25 column equilibrated in annealing buffer (10 mM MES pH 6.5 and 40 mM NaCl). DNA intended for crystallographic or biophysical experiments was subsequently concentrated in vacuo and then exchanged into annealing buffer or melting buffer (10 mM sodium phosphate pH 7.0 and 10 mM NaCl) by diafiltration (3000 nominal molecular weight limit (NMWL).

**Adduct characterization**. Absorbance and circular dichroism measurements were collected from modified oligodeoxynucleotide duplexes [5′-d(CCCCAAAGCCCG)/ d(CGGGCTTTGGGG)-3′; the underline denotes the modified nucleotide] as previously described[18]. Thermal melting profiles were also collected as previously described[18]. Melting temperatures were calculated by fitting the data to a polynomial function and determining the temperature at which the second-order derivative was equal to zero.

**AlkD/AP-DNA/DSA-Ade and AlkD/AP-DNA/CC1065-Ade crystallization**. Ternary product complexes were prepared by mixing equal volumes of 0.45 mM AlkD and 0.54 mM DSA-DNA or CC1065-DNA [5′-d(AGCAAAGGC)/d(TGCCTTTGC)-3′; the underline denotes the modified nucleotide] and incubating the mixtures at 4 °C for 30 min. Complexes were crystallized using the sitting-drop vapor-diffusion method. Drops containing DSA-DNA were prepared from 1.5 μL of protein-DNA solution [0.22 mM AlkD and 0.27 mM DNA], 0.75 μL of reservoir solution [24% (w/v) PEG 8000, 50 mM HEPES pH 7.0, and 50 mM CaCl₂], 0.75 μL of seed solution [microscopic crystals of AlkD and DNA containing an A•C mismatch], and 0.75 μL of additive solution [5% (w/v) benzamidine•HCl]. Drops containing CC1065-DNA were prepared from 1.5 μL of protein-DNA solution [0.22 mM AlkD and 0.54 mM DNA], 1.5 μL of reservoir solution [21% (w/v) PEG 8000, 50 mM HEPES pH 7.0, and 50 mM CaCl₂], and 0.75 μL of additive solution [5% (w/v) benzamidine•HCl]. All drops were equilibrated at 21 °C against 500 μL of reservoir solution. After several days, crystals were collected, briefly soaked in reservoir solution supplemented with 15% (v/v) glycerol, and flash-cooled in liquid nitrogen.

**X-ray data collection and refinement**. X-ray diffraction data were collected on beamlines 21-ID-F (λ = 0.97872 Å) and 21-ID-G (λ = 0.97857 Å) at the Advanced Photon Source. Each dataset was collected from a single cryopreserved crystal at −173 °C and processed using HKL2000[65]. Data collection statistics are provided in Supplementary Table 2. Initial phases were determined by molecular replacement, using a model of AlkD (PDB accession 3BVS)[63] placed with Phaser[66]. DNA was then manually built in Coot[67]. The entireties of the oligodeoxynucleotide duplexes were readily apparent in both complexes, as were the excised nucleobases. AlkD residues 1–225 were also clearly defined in both complexes. However, the last 12 residues (226–237) at the C terminus could not be reliably modeled. Atomic coordinates, anisotropic temperature factors (translation/libration/screw), and fractional occupancies were refined for all non-hydrogen atoms using PHENIX[68]. Hydrogen atoms were placed in riding positions and were not refined against the X-ray data. The final models were validated using MolProbity[69] and contained 96.9% of residues in the favored regions of the Ramachandran plot, 3.1% of residues in the allowed regions, and no residues in the disallowed regions. Additional refinement and validation statistics are included in Supplementary Table 2. $mF_o − DF_c$ omit maps were generated with PHENIX after removing the AP site and the modified nucleobase and performing simulated annealing on the remaining AlkD/DNA complex. Figures were prepared in PyMOL (https://www.pymol.org).

**Homology modeling**. AlkD templates were extracted from each of the three crystallographic SCPCHD product complexes. For each of the homologs, a single homology model was generated from each of the three templates using SWISS-MODEL[70]. Global quality estimates (QMEANDisCo global) for all models reached or exceeded 0.60 (0–1 scale). Local quality estimates (QMEANDisCo local) were highest for residues constituting helices in the substrate-binding cleft and lowest for residues constituting the extended loop between helices αI and αJ. Relative to AlkD, the root-mean-square deviation of all Cα positions (excluding those in the loop between helices αI and αJ) was ≤0.3 and 0.8 Å for YtkR2 and C10R5, respectively. Hypothetical product complexes were generated from the three crystallographic product complexes by replacing AlkD with the corresponding YtkR2 or C10R5 homology models. As necessary, alternate rotamers were manually selected in Coot to eliminate steric clashes or to create favorable binding interactions.

**Adduct modeling**. DSA, CC-1065, and YTM from the AlkD product complexes and ideal B-form DNA [5′-d(CCCCAAAGCCCG)/d(CGGGCTTTGGGG)-3′] generated with Coot were manually docked. Both the DNA and the adducts were maintained as rigid bodies. Placement was dictated by the geometric requirements of a covalent bond between the modification and the deoxyadenosine nucleotide and the avoidance of steric clashes with the remainder of the DNA duplex.

**Adduct excision**. Base excision was monitored by alkaline cleavage of the abasic products produced from alkylated and fluorescein (FAM)-labeled DNA [FAM-5′-d(CGGGCGGCGGCAAAGGGCGCGGGCC)/d(GGCCCGCGCCCTTTGCCGC CGCCCG)-3′; the underline denotes the modified nucleotides]. Reactions mixtures were prepared with 100 nM DNA, 20 mM Tris•HCl pH 8.0, 100 mM NaCl, 0.1 mg/ mL bovine serum albumin (BSA), and 1 mM DTT. To monitor excision under multiple- or single-turnover conditions, either 10 nM or 1 μM, respectively, of AlkD, C10R5, or YtkR2 was added. For single-turnover base excision by C10R5 mutants, which are less stable than the wild-type enzyme, reaction mixtures contained 100 nM DNA, 20 mM MOPS pH 6.8, 150 mM NaCl, and 5% (v/v) glycerol. Similarly, multiple-turnover excision by C10R5 mutants was performed in reaction mixtures containing 100 nM DNA, 20 mM MOPS pH 6.8, 200 mM NaCl, 5% (v/v) glycerol, and 0.1 mg/mL BSA. All reactions were incubated at 30 °C and quenched at different times by mixing 8 μL aliquots with 2 μL of 1 M NaOH and heating at 70 °C for 20 min. An equal volume of loading buffer [80% (w/v) formamide, 10 mM EDTA pH 8.0, 1 mg/mL bromophenol blue, and 1 mg/mL xylene cyanol] was then added before the samples were heated at 70 °C for an additional 2 min. The cleaved product was separated from the full-length substrate by denaturing urea-PAGE as previously described[43], and the fluorescence intensity of the FAM-labeled DNA was measured on a Typhoon Trio variable mode imager (GE Healthcare). Rate constants were calculated by fitting the data to a single-exponential model. For multiple-turnover experiments with AlkD, the rapid first turnover that occurred during the initial 15 s of the reaction was omitted from the fit. Experiments were performed in triplicate.

**AP-DNA incision**. AP-DNA substrate [FAM-5′-d(CGGGCGGCGGC AXXGGG CGCGGGCC)/d(GGCCCGCGCCCTTTGCCGCCGCCCG)-3′; X = A or AP] was generated by incubating 500 nM YTM-DNA, DSA-DNA, or CC1065-DNA with 1 μM AlkD, YtkR2, or C10R5 in 20 mM MOPS pH 6.8, 150 mM NaCl, 10 mM MgCl₂, 5% (v/v) glycerol, 0.1 mg/mL BSA, and 1 mM DTT at 30 °C for 3 h. THF-DNA substrate [FAM-5′-d(CGGGCGGCGGCAXAGGGGCGCGGGCC)/d(GGCCCGCGCCCTTTG CCGCCGCCCG)-3′; X = THF] was prepared by mixing THF-DNA and 3mAde nucleobase in an equal molar ratio and incubating with or without AlkD, YtkR2, or C10R5. AP-DNA incision reactions were initiated with aliquots from the base excision reaction mixtures and contained 0 or 2 nM EndoIV; 100 nM AP-DNA or THF-DNA; 200 nM AlkD, YtkR2, or C10R5; 20 mM MOPS pH 6.8; 150 mM NaCl; 10 mM MgCl₂; 5% (v/v) glycerol; 0.1 mg/mL BSA; and 1 mM DTT. Reaction mixtures were incubated at 30 °C for 1 h. Aliquots were then removed and quenched by adding an equal volume of loading buffer supplemented with 0.1 M EDTA pH 8.0 and incubating at 70 °C for 5 min. Alternatively, to determine the total fraction of AP-DNA present in the reaction mixtures, aliquots were quenched by adding 1 M NaOH to a final concentration of 0.2 M and heating at 70 °C for 20 min before adding an equal volume of loading buffer and heating at 70 °C for an additional 2 min. Samples were analyzed as described above. Experiments were performed in triplicate.

**Drug resistance**. E. coli BL21(DE3) cells were transformed with a pET27-derived plasmid encoding a fusion protein between a hexahistidine-SUMO tag and either AlkD, C10R5, or YtkR2, or encoding only the tag as a control. For growth experiments in liquid medium, precultures were grown overnight in LB medium supplemented with 0.1 mM IPTG and then diluted 100-fold in 200 μL of LB medium without IPTG but containing different concentrations of YTM, DSA, or CC-1065. Cultures were incubated in 96-well, flat-bottom plates with shaking for 18 h using a Synergy 2 multi-detector microplate reader (BioTek). Cell density was measured at 600 nm (OD₆₀₀). Experiments were performed in quadruplicate. For growth experiments on solid medium, precultures were grown overnight in LB medium supplemented with 0.5 mM IPTG. Aliquots were then diluted to 0.1 OD₆₀₀ in LB medium containing 0.1 mM IPTG and grown to an OD₆₀₀ of 1.0 to ensure that cells were in the exponential phase. New aliquots were diluted 10- to 1000-fold in LB medium without IPTG and immediately spotted on LB agar plates supplemented with various concentrations of YTM. Plates were incubated for 18 h before being imaged. Experiments were performed in duplicate. At all stages of both growth experiments, cells were grown at 37 °C and LB medium was supplemented with 30 mg/L kanamycin.

**Immunoblotting**. Portions of the precultures used for the drug resistance assays in liquid medium were diluted to an OD₆₀₀ of 0.8 in LB medium and immediately mixed with an equal volume of SDS loading buffer [100 mM Tris•HCl pH 6.8, 4% (w/v) sodium dodecyl sulfate (SDS), 4 M urea, 0.2% (w/v) bromophenol blue, 20% (v/v) glycerol, and 200 mM DTT]. Aliquots (10 μL) corresponding to half of the quantity of cells added to inoculate the drug resistance cultures were then loaded on a NuPAGE 4–12% Bis-Tris SDS-acrylamide gel (Invitrogen). Proteins in the lysate were separated by electrophoresis at 50 V for 30 min and then 100 V for 2 h before being transferred to a Trans-Blot Turbo Mini 0.2 μm nitrocellulose membrane (Bio-Rad) using a Trans-Blot Turbo transfer system (Bio-Rad). The membrane was subsequently blocked overnight with 5% (w/v) BSA in TBS buffer (20 mM Tris•HCl pH 7.6 and 150 mM NaCl) supplemented with 0.1% (v/v) Triton X-100 (TBST buffer). Excess BSA was removed by washing twice for 5 min each in TBST buffer and again for 5 min in TBS buffer. Hexahistidine-tagged protein was detected by probing the membrane at room temperature for 2 h with a monoclonal mouse anti-hexahistidine IgG1 antibody (diluted 1 : 5000; Abgent). The membrane was then washed again as described above to remove

nonspecifically bound primary antibody before being probed at room temperature for 1 h with a goat anti-mouse IgG antibody conjugated to calf intestinal alkaline phosphatase (diluted 1 : 3000; Cell Signaling Technology). Nonspecifically bound secondary antibody was removed by washing four times for 5 min each in TBST buffer. A SIGMAFAST BCIP/NBT tablet (Millipore Sigma) dissolved in 10 mL of water was then incubated with the membrane for 10 min to generate a colorimetric precipitate. The membrane was rinsed six times with water before being imaged using a GelDoc Go imaging system (Bio-Rad). Known amounts of pure hexahistidine-SUMO tag were processed in parallel with the lysate to enable determination of absolute quantities of AlkD, YtkR2, and C10R5. Experiments were performed in triplicate.

**Reporting summary**. Further information on research design is available in the Nature Research Reporting Summary linked to this article.

## Data availability

Atomic coordinates and structure factors generated in this study were deposited in the Protein Data Bank under accession codes 7LXJ (AlkD/AP-DNA/DSA-Ade) and 7LXH (AlkD/AP-DNA/CC1065-Ade). Previously published structures of AlkD, AlkD/AP-DNA/YTM-Ade, and DSA-DNA are available in the Protein Data Bank under accession codes 3BVS, 5UUG, and 1DSA, respectively. Source data are provided with this paper.

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

## Acknowledgements

This work was funded by the National Science Foundation (MCB-1928918 to B.F.E.) and the National Institutes of Health (R01 CA208669 to D.L.B.). Use of the Advanced Photon Source, an Office of Science User Facility operated for the U.S. Department of Energy Office of Science by Argonne National Laboratory, was supported by the U.S. Department of Energy (DE-AC02-06CH11357). Use of LS-CAT Sector 21 was supported by the Michigan Economic Development Corporation and the Michigan Technology Tri-Corridor (085P1000817).

## Author contributions

E.A.M. and B.F.E. conceived the project. E.A.M., J.D., and B.F.E. designed experiments. E.A.M. performed biophysical and structural experiments. J.D. performed cellular experiments. E.A.M. and J.D. performed biochemical experiments. G.-L.T. and D.L.B. provided reagents. E.A.M., J.D., and B.F.E. analyzed data and wrote the manuscript.

## Competing interests

The authors declare no competing interests.
