## [Peer Review File · Nature Communications]

REVIEWER COMMENTS

Reviewer #1 (Remarks to the Author):

The manuscript submitted by Mullins and coworkers, describes a comprehensive biochemical and structural analysis of AlkD-family enzymes that likely evolved to protect against duocarmycin family natural products that exert cytotoxicity by forming adducts into the minor groove of DNA. Several interesting observations emerged from this systematic study. Here it is shown that the AlkD-resistance gene from the biosynthetic gene cluster shows enhanced activity to protect against the specific antibiotic generated by the gene cluster. Furthermore, rather than enhanced binding affinity, a common feature among the enzymes tested is that greater protection is provided by enhanced multiple turnover removal of the adduct which is provided by weakened product interactions. Finally, the properties of AlkB from *B. cereus* which does not produce a duocarmycin antibiotic, but may encounter environmental sources of these DNA alkylating agents, is very efficient at recognizing and excising structurally distinct alkylating agents. This comes at the cost of higher product inhibition and reduced multiple turnover removal of adducts. As the authors point out, this makes evolutionary sense that producers may need to deal with higher levels of DNA adducts than organisms that may coexist in the same environment. The quality of the experiments are high and the presentation is excellent, and I expect that this could be a definitive publication in establishing the structure/function relationship amongst these glycosylases that provide resistance against minor-groove binding DNA alkylating agents.

Specific comments.

1. It would be very interesting to characterize the single turnover glycosylase activity of AlkD and YrkR2. Minimally, it should be shown that the chemical quench is effective in stopping the glycosylase activity of AlkD and ThkR2. It would be most informative to use rapid mixing approaches to define the rate constant for the glycosylase activity.
2. This is a related point, but it bears repeating that it is very important to compare the affinity of the AlkD family members for the respective duocarmycin DNA adducts. If single turnover kinetics are employed, then the concentration of enzyme should be varied. If multiple turnover kinetics are employed, then the concentration of DNA substrate should be varied in order to compare the catalytic efficiency of each enzyme. With the presented data, it is not addressed whether the resistance enzymes are capable of both efficient recognition and high turnover.
3. It would be interesting to have a little more discussion of the biological results. There may be a profound point here if the authors are correct that AlkD can efficiently recognize YTM, but simply fails to release the product in vivo. Is the problem that there is a molar excess of adduct over AlkD

protein, or is the AlkD product complex an impediment to growth? Some estimate of the stoichiometry might be informative.

Reviewer #2 (Remarks to the Author):

The Mullins et al manuscript investigates the resistance mechanism of duocarmycin family metabolites. Some bacteria produce secondary metabolites that damage DNA providing a selective growth advantage in their niche. A new frontier in microbiology is understanding how secondary metabolites damage DNA and how that damage is mitigated by the organism producing the genotoxin. This work specifically investigates the YtkR2 and C10R5 two uncharacterized AlkD family glycosylases present in the biosynthetic cluster producing yatakemycin and CC-1065. Overall, the study is very well done, clearly important and it provides a significant mechanistic step forward in understanding bacterial resistance to naturally produced clastogens.

Major:

Figure 3 is not very intuitive and this experimental approach is not the appropriate method to demonstrate the point. When you measure OD, you are measuring live and dead cells. Therefore, as opposed to measuring lagtime by OD, the authors should measure the end point of colony forming units (CFUs). If you plate for survival what is the result? If these duocarmycin metabolites are as toxic as the mechanism suggests then the authors should present a killing curve where they either vary concentration or time and plate for survivors. Alternatively, the authors could use a chronic dose and perform spot titer dilutions on the respective metabolite and determine if the cognate glycosylase provides resistance in that scenario. Finally, BL21(DE3) is simply not the appropriate genetic background for this type of experiment which is compounded by expressing tagged proteins. Just place the genes (without tags) on a normal vector under an E. coli constitutive promoter (or inducible if necessary) and transform a more appropriate genetic background such as MC1061 or MG1655.

Minor:

Page 8-9 homology model section: The modeling seems well done. The issue here is that the authors are describing structural features which are modeled and therefore predicted and not known as structural elements. The authors should tighten up the language in this section for which portions represent modeled elements and which are known to be structural elements.

Title: I recommend removing "duorcarmycin family" from the title and changing it to "Structural evolution of a DNA repair self-resistance mechanism targeting secondary metabolites" Most people may not be familiar with the duorcarmycin family, and thus removing this gives the paper a broader appeal.

page 6 second paragraph line one "why" should be "how"

Reviewer #3 (Remarks to the Author):

The manuscript by Mullins et al investigates the molecular mechanism of microbes self-resistance to three DNA alkylating genotoxins that belong to the spirocyclopropylcyclohexadienone (SCPCHD) family: yatakemycin (YTM), CC-1065, and DSA. The authors explore how the SCPCHD-specific glycosylases (YtkR2 and C10R5) have evolved to provide better resistance than the glycosylase serving in the general DNA repair pathway (AlkD).

The authors kinetically and structurally characterize the ability of AlkD, YtkR2, and C10R5 to repair DNA lesions caused by binding to the SCPCHD toxins, YTM, CC-1065, and DSA. In their study, through molecular homology modeling, they proposed that binding with SCPCHD adducts creates an extended network of non-covalent CH- π interactions with both DNA strands, greatly stabilizing the DNA duplex. This DNA duplex stabilization via non-covalent interactions has also been suggested in previously published papers by their group, in the YTM bound DNA duplex. Furthermore, they infer that the faster catalytic rate of the glycosylases YtkR2 and C10R5, at steady state in vitro, are responsible for conferring enhanced cellular resistance against YTM, CC-1065, and DSA in vivo. To further address the mechanism, structural studies and molecular homology modeling were performed by the authors to conclude that reduced product binding affinity in YtkR2 or C10R5 contributes to their ability to remove the YTM/CC1065-Ade more efficiently than AlkD.

This study expanded the substrate spectrum of non-base-flipping DNA glycosylases and broadens the biological utility of the BER pathway. However, some of the conclusions are not sufficiently supported. Several further experiments are necessary before publication.

Major points:

1) The authors claim that substrate specificity is driven almost entirely by the rate of product release and that product inhibition is the major contributor to specificity to these enzymes. While their previous work (Mullins et al Nat Chem Bio 17) beautifully shows this for AlkD, it is not convincingly shown here for YtkR2 and C10R5. While it is likely that this is the case for YtkR2 and C10R5, the authors have only shown that the rates of catalysis are drastically different at varying enzyme concentration; there are numerous other explanations for this phenomenon. In their previous work, the authors could show that addition of exogenous product could inhibit AlkD. Can the authors perform similar experiments here for YTM or CC1065? If not, some other means of convincingly showing that product inhibition is occurring is necessary.

2) The authors claim that specificity for these enzymes is dictated by increasing the product release rate, as discussed above. However, the authors explore substrate specificity by making a chimeric version of C10R5 enzyme with mutations to recapitulate the YtkR2 adduct binding site. However, they only test these mutants in the single turnover assay that doesn't probe product release. To confirm the biological relevance of the mutations, a cell growth assay needs to be performed with the mutants that shows improved resistance to YTM, or perform the kinetics experiments using multiple turnover conditions. Ideally both, but one set of assays should suffice.

3) The protein expression level for AlkD, YtkR2, and C10R5 in the E. coli strains should be tested to exclude the possibility that the observed divergent resistances provided by the three enzymes are not due to (potentially) differing protein expression levels.

4) The generation of the homology models are not adequately described, nor the confidence in the resulting models. How were the models generated? Did the authors only use the top model? How did the top 5 models look relative to each other, particularly in the active site and adduct binding sites? Ca RMSD from AlkD?

Minor points:

1) In Figure 1b, the labels on top of the arrows are confusing. The scheme does not represent the full repair steps, but rather just base excision. I'm also not sure that fig 1c is necessary.

2) In Figure 4b, what is the doublet band detected in the DSA adduct excision product?

3) In the cell growth assay, some references should be cited for the quantification method.

REVIEWER COMMENTS AND RESPONSES

We thank the reviewers for their careful consideration of the manuscript. Upon their requests, we performed several additional experiments, which we describe below. We believe these new results greatly strengthen this study.

Reviewer #1: (Remarks to the Author)

The manuscript submitted by Mullins and coworkers, describes a comprehensive biochemical and structural analysis of AlkD-family enzymes that likely evolved to protect against duocarmycin family natural products that exert cytotoxicity by forming adducts into the minor groove of DNA. Several interesting observations emerged from this systematic study. Here it is shown that the AlkD-resistance gene from the biosynthetic gene cluster shows enhanced activity to protect against the specific antibiotic generated by the gene cluster. Furthermore, rather than enhanced binding affinity, a common feature among the enzymes tested is that greater protection is provided by enhanced multiple turnover removal of the adduct which is provided by weakened product interactions. Finally, the properties of AlkB from *B. cereus* which does not produce a duocarmycin antibiotic, but may encounter environmental sources of these DNA alkylating agents, is very efficient at recognizing and excising structurally distinct alkylating agents. This comes at the cost of higher product inhibition and reduced multiple turnover removal of adducts. As the authors point out, this makes evolutionary sense that producers may need to deal with higher levels of DNA adducts than organisms that may coexist in the same environment. The quality of the experiments are high and the presentation is excellent, and I expect that this could be a definitive publication in establishing the structure/function relationship amongst these glycosylases that provide resistance against minor-groove binding DNA alkylating agents.

Specific comments:

1. It would be very interesting to characterize the single turnover glycosylase activity of AlkD and YrkR2. Minimally, it should be shown that the chemical quench is effective in stopping the glycosylase activity of AlkD and ThkR2. It would be most informative to use rapid mixing approaches to define the rate constant for the glycosylase activity.

We confirmed that the chemical quench is effective at stopping glycosylase activity by adding the sodium hydroxide quench prior to initiating the reaction with enzyme. Under these modified conditions, we observed no activity with any of the homologs, indicating the quench is both rapid and complete. These results were added to Supplementary Fig. 5.

2. This is a related point, but it bears repeating that it is very important to compare the affinity of the AlkD family members for the respective duocarmycin DNA adducts. If single turnover kinetics are employed, then the concentration of enzyme should be varied. If multiple turnover kinetics are employed, then the concentration of DNA substrate should be varied in order to compare the catalytic efficiency of each enzyme. With the presented data, it is not addressed whether the resistance enzymes are capable of both efficient recognition and high turnover.

We agree that a detailed kinetic analysis of the type suggested by the reviewer would provide a more quantitative understanding of substrate recognition. However, given the scope (three enzymes acting on each of three substrates, for a total of nine combinations, each with varying amount of enzyme or substrate), this would constitute an entire study of its own and would be better suited for a subsequent publication. Our single-turnover excision experiments clearly show that all of the homologs are capable of high turnover, and strongly suggest that

inefficient recognition slows excision of YTM adducts by C10R5. Our homology-based mutational analysis, which has been expanded in the revised manuscript, demonstrates that this inefficient recognition can be substantially improved by simply eliminating steric clashes, without introducing additional adduct-specific binding interactions. This is consistent with our understanding of the predominantly non-specific substrate recognition mechanism used by these enzymes. Furthermore, we are able to explain the trends in cellular resistance that we observe through tight binding of the abasic DNA products, as suggested by our multiple-turnover excision experiments. In the revised manuscript, we have further investigated the role of product affinity through AP endonuclease assays. These new data were added as Figs. 4e and 6e,f and Supplementary Figs. 7, 11, and 12 in the revised manuscript.

3. It would be interesting to have a little more discussion of the biological results. There may be a profound point here if the authors are correct that AlkD can efficiently recognize YTM, but simply fails to release the product in vivo. Is the problem that there is a molar excess of adduct over AlkD protein, or is the AlkD product complex an impediment to growth? Some estimate of the stoichiometry might be informative.

We added additional discussion of possible biological mechanisms by which tight product binding might limit cellular resistance. To enhance this analysis, we performed additional experiments that provide further insight into the deleterious effects of high product affinity, specifically how binding of abasic products by the three homologs impedes the subsequent step in the base excision repair pathway. The results from these experiments were added as Fig. 4e and Supplementary Fig. 7. In an attempt to determine stoichiometry, we experimentally determined the protein copy number per colony forming unit (CFU) by Western blot analysis of expressed protein and counting of the number of colony forming units per OD. However, we are unable to determine the fraction of the drug molecules that form adducts in the DNA, making any estimate of protein/adduct stoichiometry extremely uncertain and potentially misleading. Therefore, we are hesitant to discuss specific protein/adduct stoichiometries in our experiments.

Reviewer #2 (Remarks to the Author):

The Mullins et al manuscript investigates the resistance mechanism of duocarmycin family metabolites. Some bacteria produce secondary metabolites that damage DNA providing a selective growth advantage in their niche. A new frontier in microbiology is understanding how secondary metabolites damage DNA and how that damage is mitigated by the organism producing the genotoxin. This work specifically investigates the YtkR2 and C10R5 two uncharacterized AlkD family glycosylases present in the biosynthetic cluster producing yatakemycin and CC-1065. Overall, the study is very well done, clearly important and it provides a significant mechanistic step forward in understanding bacterial resistance to naturally produced clastogens.

Major:

Figure 3 is not very intuitive and this experimental approach is not the appropriate method to demonstrate the point. When you measure OD, you are measuring live and dead cells. Therefore, as opposed to measuring lagtime by OD, the authors should measure the end point of colony forming units (CFUs). If you plate for survival what is the result? If these duocarmycin metabolites are as toxic as the mechanism suggests then the authors should present a killing curve where they either vary concentration or time and plate for survivors. Alternatively, the authors could use a chronic dose and perform spot titer dilutions on the respective metabolite

and determine if the cognate glycosylase provides resistance in that scenario. Finally, BL21(DE3) is simply not the appropriate genetic background for this type of experiment which is compounded by expressing tagged proteins. Just place the genes (without tags) on a normal vector under an E. coli constitutive promoter (or inducible if necessary) and transform a more appropriate genetic background such as MC1061 or MG1655.

Measuring the lag time of cell growth is a well-established method to assess toxicity of antibiotics on bacteria, and we have added multiple references that support our use of this methodology [Fridman, et al. (2014) Optimization of lag time underlies antibiotic tolerance in evolved bacterial population. *Nature*, 513: 418-421; Li, et al. (2016) The importance of lag time extension in determining bacterial resistance to antibiotics. *Analyst*, 141: 3059-3067].

We agree that cell survival is an informative measure of toxicity. However, we are interested in overall growth inhibition as opposed to only cell death. While measuring toxicity as a function of lag time does not differentiate between live cells not dividing and dead cells not dividing, it does demonstrate the relative resistance against each SCPCHD compound conferred by each homolog, which is the aim of our study. The plots showing lag time versus drug concentration in Fig. 3 are an efficient and clear manner to summarize the 72 growth curves in Supplementary Fig. 3. Moreover, we previously have shown a strong correlation between results obtained from growth curves and the type of spot titer assays suggested by the reviewer [Mullins et al. (2015) *Nature*, 527: 254-258; Mullins et al. (2017) *Nat Chem Biol*, 13: 1002-1008], including new data presented in Fig. 6f and Supplementary Fig. 12 of the revised manuscript.

We respectfully disagree that BL21(DE3) cells are an inappropriate genetic background for these experiments. BL21(DE3) have been used in a number of prior toxicity studies [e.g., Liu et al. (2008) Adrimid producers encode an acetyl-CoA carboxyltransferase subunit resistant to the action of the antibiotic. *Proc Natl Acad Sci USA*, 105: 13321-13326; Biggins et al. (2003) Resistance to enediyne antitumor antibiotics by CalC self-sacrifice. *Science*, 301: 1537-1541]. Additionally, as our experiments are comparative in nature, and all of the homologs are being characterized with the same genetic background, any possible deficiencies would be normalized and thus unlikely to affect our analysis of relative resistance. The same is true for the affinity tag, which is present on all of the homologs (and expressed alone in the control experiments). The affinity tag also provided the benefit of allowing for quantitation of protein expression using an α -hexahistidine antibody.

Minor:

Page 8-9 homology model section: The modeling seems well done. The issue here is that the authors are describing structural features which are modeled and therefore predicted and not known as structural elements. The authors should tighten up the language in this section for which portions represent modeled elements and which are known to be structural elements.

We altered our language in this section to further emphasize which of our structural observations are based on predictive models.

Title: I recommend removing "duorcarmycin family" from the title and changing it to "Structural evolution of a DNA repair self-resistance mechanism targeting secondary metabolites" Most people may not be familiar with the duorcarmycin family, and thus removing this gives the paper a broader appeal.

As suggested, we modified the title to broaden the paper's appeal.

page 6 second paragraph line one "why" should be "how"

This has been corrected.

Reviewer #3 (Remarks to the Author):

The manuscript by Mullins et al investigates the molecular mechanism of microbes self-resistance to three DNA alkylating genotoxins that belong to the spirocyclopropylcyclohexadienone (SCPCHD) family: yatakemycin (YTM), CC-1065, and DSA. The authors explore how the SCPCHD-specific glycosylases (YtkR2 and C10R5) have evolved to provide better resistance than the glycosylase serving in the general DNA repair pathway (AlkD).

The authors kinetically and structurally characterize the ability of AlkD, YtkR2, and C10R5 to repair DNA lesions caused by binding to the SCPCHD toxins, YTM, CC-1065, and DSA. In their study, through molecular homology modeling, they proposed that binding with SCPCHD adducts creates an extended network of non-covalent CH- π interactions with both DNA strands, greatly stabilizing the DNA duplex. This DNA duplex stabilization via non-covalent interactions has also been suggested in previously published papers by their group, in the YTM bound DNA duplex. Furthermore, they infer that the faster catalytic rate of the glycosylases YtkR2 and C10R5, at steady state in vitro, are responsible for conferring enhanced cellular resistance against YTM, CC-1065, and DSA in vivo. To further address the mechanism, structural studies and molecular homology modeling were performed by the authors to conclude that reduced product binding affinity in YtkR2 or C10R5 contributes to their ability to remove the YTM/CC1065-Ade more efficiently than AlkD.

This study expanded the substrate spectrum of non-base-flipping DNA glycosylases and broadens the biological utility of the BER pathway. However, some of the conclusions are not sufficiently supported. Several further experiments are necessary before publication.

Major points:

1) The authors claim that substrate specificity is driven almost entirely by the rate of product release and that product inhibition is the major contributor to specificity to these enzymes. While their previous work (Mullins et al Nat Chem Bio 17) beautifully shows this for AlkD, it is not convincingly shown here for YtkR2 and C10R5. While it is likely that this is the case for YtkR2 and C10R5, the authors have only shown that the rates of catalysis are drastically different at varying enzyme concentration; there are numerous other explanations for this phenomenon. In their previous work, the authors could show that addition of exogenous product could inhibit AlkD. Can the authors perform similar experiments here for YTM or CC1065? If not, some other means of convincingly showing that product inhibition is occurring is necessary.

Similar to our previously published experiments with AlkD, we have now examined the effect of each homolog on the AP endonuclease activity of EndoIV. With AP-DNA generated from each of the SCPCHD adducts, we observed a clear difference between the rates at which AP-DNA was incised in the presence of AlkD and the rates at which AP-DNA was incised in the presence of YtkR2 or C10R5. These findings further support our conclusion that YtkR2 and C10R5 possess relatively low product affinities. Additionally, the demonstration of inhibition of AP endonuclease activity in vitro has implications for cellular resistance, which

we now discuss in the text. The results from these experiments have been added as Fig. 4e and Supplementary Fig. 7.

2) The authors claim that specificity for these enzymes is dictated by increasing the product release rate, as discussed above. However, the authors explore substrate specificity by making a chimeric version of C10R5 enzyme with mutations to recapitulate the YtkR2 adduct binding site. However, they only test these mutants in the single turnover assay that doesn't probe product release. To confirm the biological relevance of the mutations, a cell growth assay needs to be performed with the mutants that shows improved resistance to YTM, or perform the kinetics experiments using multiple turnover conditions. Ideally both, but one set of assays should suffice.

We agree that this is an important point and performed both sets of experiments. We observed an increased rate of YTM-Ade excision with all C10R5 mutants under multiple-turnover conditions and a corresponding increase in cellular resistance in spot assays. These results confirm the biological relevance of these mutations, and validate the homology models used to choose them. The results from the multiple-turnover excision experiments have been added as Fig. 6e and Supplementary Fig. 11, and the spot assays are shown in Fig. 6f and Supplementary Fig. 12.

3) The protein expression level for AlkD, YtkR2, and C10R5 in the E. coli strains should be tested to exclude the possibility that the observed divergent resistances provided by the three enzymes are not due to (potentially) differing protein expression levels.

We quantified protein expression by Western blot using the hexahistidine tag present on each homolog as an epitope. Expression of the homologs were within 2.5-fold of one another, and these small differences do not correlate with differences in cellular resistance. We have added these results to Supplementary Fig. 3.

4) The generation of the homology models are not adequately described, nor the confidence in the resulting models. How were the models generated? Did the authors only use the top model? How did the top 5 models look relative to each other, particularly in the active site and adduct binding sites? Ca RMSD from AlkD?

We added a description of homology model generation to the Methods.

Minor points:

1) In Figure 1b, the labels on top of the arrows are confusing. The scheme does not represent the full repair steps, but rather just base excision. I'm also not sure that fig 1c is necessary.

We changed "Damage" and "Repair" to "Modification" and "Excision" in Fig. 1b.

The presence of *ytkR2* and *c10R5* within the biosynthetic gene clusters of YTM and CC-1065 is the basis for this study. As such, we prefer to retain Fig. 1c to emphasize this point.

2) In Figure 4b, what is the doublet band detected in the DSA adduct excision product?

Each product band corresponds to modification of the 5'...CAAAG...3' sequence at a different adenosine nucleotide. As indicated in Fig. 4a, DSA predominantly modifies the A at the 3'-end of the sequence, producing the more intense product band. Less frequently, DSA

modifies the central A instead, producing the less intense product band. We added an explanation of this to the legend for Fig. 4b.

3) In the cell growth assay, some references should be cited for the quantification method.

We added two references that explain the utility of measuring lagtime as a means of assessing toxicity.

REVIEWERS' COMMENTS

Reviewer #1 (Remarks to the Author):

The revised manuscript by Mullins and colleagues includes new experimental data and has carefully addressed each of the reviewers' critiques/suggestions. These revisions have significantly improved the manuscript. However, in reading the revised manuscript there are a few new points introduced by the revisions that the authors may wish to address prior to publication.

1. The new experiment in Figure 4e is compelling, in which accessibility of the abasic product by EndoIV was tested. In my opinion the bar graphs are confusing and not the cleanest way to introduce the idea of protection to a general audience. Consider defining fractional or percent protection from endonuclease incision (or just call it protection of the abasic site and explain what that is in the legend). Fraction protected = $1 - (\text{fraction cleaved by endoIV} / \text{fraction cleaved by NaOH})$; multiple by 100 to get percent protection. For transparency, a summary table and calculation could be included in the supplement along with Supplemental Figure 7.

2. In Figure 6d,e it is notable that the glycosylase activity increased with the mutations. However, I don't agree with the description and conclusion on page 10 and 11. These improved glycosylases do not apparently show improved protection in vivo. This is not something to be ignored. It could mean that they are expressed more poorly than the wild-type YtkR2 and thus improved activity is counteracted by reduced enzyme. Alternatively, it could mean that activity per se is not sufficient for protection. Could it be that locating the sites of alkylation is more important than rate of excision? Could product inhibition be important? It would be interesting to get to the bottom of this, but there are a lot of conclusions that can be drawn from the current work, and this point could be left as a point of discussion rather than being experimentally resolved.

Reviewer #2 (Remarks to the Author):

The manuscript is much improved.

Reviewer #3 (Remarks to the Author):

The manuscript by Mullins et al is substantially improved. The authors have adequately addressed all of my concerns, and (IMHO) the concerns of the other reviewers as well. In particular, the manuscript has greatly benefitted by the inclusion of multiple turnover kinetics of variant enzymes and expansion of the studies in cells. I support immediate publication of the manuscript.

REVIEWER COMMENTS AND RESPONSES #2

Reviewer #1 (Remarks to the Author):

The revised manuscript by Mullins and colleagues includes new experimental data and has carefully addressed each of the reviewers' critiques/suggestions. These revisions have significantly improved the manuscript. However, in reading the revised manuscript there are a few new points introduced by the revisions that the authors may wish to address prior to publication.

1. The new experiment in Figure 4e is compelling, in which accessibility of the abasic product by EndoIV was tested. In my opinion the bar graphs are confusing and not the cleanest way to introduce the idea of protection to a general audience. Consider defining fractional or percent protection from endonuclease incision (or just call it protection of the abasic site and explain what that is in the legend). Fraction protected = $1 - (\text{fraction cleaved by endoIV} / \text{fraction cleaved by NaOH})$; multiple by 100 to get percent protection. For transparency, a summary table and calculation could be included in the supplement along with Supplemental Figure 7.

We thank the reviewer for this suggestion. However, after careful consideration, we believe this alternative manner of presentation is problematic, for two reasons. First, the calculation proposed to determine the percent of abasic DNA protected from enzymatic cleavage does not apply to THF-DNA. Unlike AP-DNA, THF-DNA is not cleaved by mild treatment with NaOH. While we know the percent of abasic DNA in samples containing THF-DNA is 100%, using this value in the suggested equation indicates ~17% of THF-DNA is protected in the mock reaction. Obviously, in the absence of a DNA glycosylase homolog, the percent of abasic DNA protected is 0%. Second, reaction conditions were chosen such that the rapidly cleaved substrates would not reach the endpoint of the reaction within the time period of the assay, which would have allowed the slowly cleaved substrates to "catch up." As such, the suggested equation overestimates the percent of abasic DNA protected, as noted in the first point regarding THF-DNA. For these reasons, we prefer to retain the original presentation of the AP endonuclease data.

2. In Figure 6d,e it is notable that the glycosylase activity increased with the mutations. However, I don't agree with the description and conclusion on page 10 and 11. These improved glycosylases do not apparently show improved protection in vivo. This is not something to be ignored. It could mean that they are expressed more poorly than the wild-type YtkR2 and thus improved activity is counteracted by reduced enzyme. Alternatively, it could mean that activity per se is not sufficient for protection. Could it be that locating the sites of alkylation is more important than rate of excision? Could product inhibition be important? It would be interesting to get to the bottom of this, but there are a lot of conclusions that can be drawn from the current work, and this point could be left as a point of discussion rather than being experimentally resolved.

Because the apparent differences in YTM resistance in the spot-titer experiments are modest, we performed a densitometric analysis of cell growth (see below) to more quantitatively demonstrate the resistance conferred by each C10R5 mutant relative to the resistance conferred by wild-type C10R5. This quantitation shows that the C10R5 F22T-S25G double mutant does provide additional cellular resistance to YTM, consistent with our original conclusion. In contrast, the data for the C10R5 F22T-S25G-A40R triple mutant are less clear. Thus, we have revised our language in the Results and have added a brief discussion of factors that may limit the resistance conferred by these mutants.

Protection of cells from YTM toxicity by expression of C10R5 double (left) and triple (right) mutants. C10R5 [WT, double mutant (DM), triple mutant (TM)] cell growth shown in Supplementary Figure 12 was quantified by densitometry using ImageJ. Relative cell growth plotted on the Y-axis is defined as $I_{mut}/(I_{WT}+I_{mut})$, where I_{mut} and I_{WT} are the integrated intensities of C10R5 mutants and C10R5 wild-type, respectively. A value of 0.5 (dashed line) denotes no difference between mutant and wild-type.

Reviewer #2 (Remarks to the Author):

The manuscript is much improved.

Reviewer #3 (Remarks to the Author):

The manuscript by Mullins et al is substantially improved. The authors have adequately addressed all of my concerns, and (IMHO) the concerns of the other reviewers as well. In particular, the manuscript has greatly benefitted by the inclusion of multiple turnover kinetics of variant enzymes and expansion of the studies in cells. I support immediate publication of the manuscript.